# Impact of Stress during COVID-19 Pandemic in Italy: A Study on Dispositional and Behavioral Dimensions for Supporting Evidence-Based Targeted Strategies

**DOI:** 10.3390/ijerph21030330

**Published:** 2024-03-12

**Authors:** Giuseppe Alessio Platania, Simone Varrasi, Claudia Savia Guerrera, Francesco Maria Boccaccio, Vittoria Torre, Venera Francesca Vezzosi, Concetta Pirrone, Sabrina Castellano

**Affiliations:** 1Department of Educational Sciences, University of Catania, 95124 Catania, Italy; giuseppe.platania@phd.unict.it (G.A.P.); claudia.guerrera@phd.unict.it (C.S.G.); francesco.boccaccio@phd.unict.it (F.M.B.); vittoria.torre@phd.unict.it (V.T.); concetta.pirrone@unict.it (C.P.); sabrina.castellano@unict.it (S.C.); 2Unità Operativa Complessa di Neuropsichiatria Infantile, Presidio Ospedaliero di Acireale, ASP 3 Catania, 95024 Acireale, Italy; veravezzosi@libero.it

**Keywords:** COVID-19, stress, personality factors, intolerance to uncertainty, coping strategies, interventions

## Abstract

The COVID-19 pandemic caused critical mental health issues and lifestyle disruptions. The aim of this study was to explore, during the lockdown of second-wave contagions in Italy, how stress was affected by dispositional (personality factors and intolerance to uncertainty) and behavioral (coping strategies) dimensions, how these variables differed among sex, age, educational, professional, and health groups, and how the various changes in work and daily routine intervened in the psychological impact of the emergency. Our results highlight that women, the youngs, students/trainees, those with chronic diseases, those who stopped their jobs due to restrictions, and those who left home less than twice a week were more stressed, while health professionals showed lower levels of the same construct. Those with higher levels of stress used more coping strategies based on avoidance, which positively correlated with age, agreeableness, conscientiousness, and intolerance to uncertainty, and negatively with openness. Stress levels also positively correlated with agreeableness, conscientiousness, intolerance to uncertainty, and seeking of social support, and negatively with openness, a positive attitude, and a transcendent orientation. Finally, stress was predicted mainly by behavioral dimensions. Our results are discussed and framed within the literature, as important insights for targeted intervention strategies to promote health even in emergencies.

## 1. Introduction

In January 2020, the World Health Organization (WHO) declared a state of a “public health emergency of international concern” due to the rapid global spread of COVID-19, an infectious disease caused by the SARS-CoV-2 virus. The transmission of the virus occurred through respiratory droplets released by infected individuals when coughing or sneezing. Common symptoms included fever, fatigue, and a persistent dry cough. Older individuals and those with pre-existing health conditions were at a higher risk of experiencing severe symptoms and death, while children were less likely to develop concerning clinical conditions [1].

To mitigate the spread of SARS-CoV-2, governments worldwide implemented public health measures such as stay-at-home orders, frequent handwashing, the maintenance of physical distance, and the wearing of face masks [2]. Although these measures were effective in preventing the further spread of SARS-CoV-2 [3], they disrupted daily routines [4] with an impact on economic, social, and psychological wellbeing [5].

Indeed, during the COVID-19 pandemic there was an increase in perceived stress, depressive symptoms, anxiety, and sleep disturbances, particularly in students, women, young people, and healthcare professionals, with medium- and long-term effects [6,7,8].

Moreover, several studies confirmed increased levels of peritraumatic stress compared to the pre-pandemic period [9,10]. Potential stress factors were represented by a longer quarantine, fear of infection, frustration, boredom, inadequate information, economic loss, and stigma [11,12]. It should be taken in mind, however, that stress is a natural reaction for promoting the successful management of challenging stimuli. As a consequence, stress may be functional within certain limits. In the case of the pandemic, instead, the constant tension experienced by some individuals combined with other specific factors often led to the development of detrimental consequences.

In particular, five factors contributed to the COVID-19 stress syndrome: fear of the dangerousness of the disease, fear of its socio-economic costs, fear that foreigners could spread SARS-CoV-2, symptoms of traumatic stress associated with direct or vicarious exposure to COVID-19, and compulsive control related to COVID-19. The pre-existence of psychopathology or excessive avoidance or a panic disorder, and coping difficulties during self-isolation linked to COVID-19 were correlated with the severity of the syndrome [13].

In addition to this condition, restrictive social and public health measures, changes in night sleep, the use of psychotropic drugs, the search for psychological help, and behavioral disengagement contributed to moderate or worsened anxiety symptoms, like those of generalized anxiety disorder [14,15]. In this scenario, stress related to the pandemic was affected by both dispositional and behavioral dimensions, and namely by variables like personality factors, intolerance to uncertainty, and coping strategies [16,17,18].

Indeed, the Big Five personality traits played a key role in the management of stressful events. More in detail, mental well-being and loneliness during the pandemic were positively associated with neuroticism and extraversion [19,20]. In particular, it was shown that individuals with a personality profile characterized by high neuroticism, low extroversion, low agreeableness, low conscientiousness, and low openness had higher levels of COVID-19 fear, perceived stress, and poor sleep quality compared to those with adaptive personality profiles (moderate levels of neuroticism, extroversion, agreeableness, conscientiousness, and openness) and highly adaptive ones (low neuroticism, high extroversion, high agreeableness, high conscientiousness, and high openness) [21].

Moreover, the key factor in maintaining stress and a generalized anxiety disorder was the intolerance to uncertainty [22,23], that is, the dispositional tendency of the individual to think of encountering a threatening event regardless of its occurrence [24]. The overall uncertainty caused by the COVID-19 pandemic resulted in the inability to control or predict events related to the pandemic [25]. This was one of the most stressful aspects and, along with misinformation and distorted news, it affected mental health [26,27].

Additional factors affecting the management of stressful situations such as COVID-19 emergencies were coping strategies. They are defined as “cognitive and behavioral efforts to dominate, reduce or tolerate internal and/or external demands that are created by the stressful transaction” [28]. Analyzing scientific evidence, it emerged that positive coping was the most widely used style in the pandemic, in particular acceptance, planning, and transcendent orientation, while avoidant coping strategies were used by the weakest categories of the population and were related to higher psychopathological levels, in particular anxious and depressive symptoms [29].

Although those dispositional and behavioral dimensions were shared by the entire population, each demographic group showed a different impact of the pandemic based on economic, social, professional, family, and routine aspects [30,31,32,33,34,35].

But why still talk of the COVID-19 pandemic? Though nowadays it seems that the COVID-19 disease has been almost forgotten, we must not ignore that the pandemic had a tremendous negative impact on the population’s physical and especially psychological health. Common experience and the literature show that in a world where stress is the social disease of the century, COVID-19 was able to critically worsen its impact unevenly, burdening some individuals more than others [14,36,37]. We believe that this traumatic experience can help us understand what the most fragile segments of the population are, and therefore which of them should be promptly supported in case of emergencies. Moreover, data collected in that dramatic period can shed light on the dispositional and behavioral factors affecting stress during emergencies, so that supportive interventions can be designed appropriately. From this point of view, our work intends to move forward the evidence shown by the literature so far [38,39,40].

The present study, therefore, aimed at exploring the different risk profiles of the Italian population during the lockdown of second-wave contagions. More in detail, we explored how stress levels were affected by dispositional (personality factors and intolerance to uncertainty) and behavioral (coping strategies) dimensions, and how those variables differed between several demographic groups (sex, age, level of education, occupation, health status, work, and daily routine changes). Therefore, the objectives of the present study were: (1) to verify the relationships between stress, personality factors, intolerance to uncertainty, and coping strategies during the lockdown of second-wave contagions in Italy; (2) to understand how categories of the population have been differently affected; and (3) to explore whether stress levels were predicted by either dispositional or behavioral factors. Finally, targeted interventions to improve the quality of life of the population are hypothesized.

## 2. Materials and Methods

### 2.1. Ethical Considerations

The study was designed in full compliance with the Declaration of Helsinki. Each participant, who volunteered to join the study, accepted a written informed consent before starting the collection of data. The informed consent included the reasons for the study, responsibilities and information about data use, anonymity, and a data protection clause. The questionnaires were fully anonymous and administered online. The collected data were analyzed statistically and in an aggregate form. The study was approved by the Institutional Review Board of the Department of Educational Sciences, University of Catania, with the number Ierb-Edunict-2023.11.21/02, 21 November 2023.

### 2.2. Participants

The sample included 550 individuals, recruited using non-probabilistic snowball sampling. It consisted of 142 men (25.8%) and 408 women (74.2%), whose mean age was about 32 years (M = 31.98, SD = 11.69, minimum = 18, maximum = 73). Of them, 5.5% had a middle school license, 42.4% completed high school, 46% held a university degree, and 6.2% attended a postgraduate course. Given the professional heterogeneity of the sample and the scope of the study, we decided to subdivide participants according to their job. Recruited subjects worked as health professionals (12.5%), students/trainees (32%), unemployed/housewives/retirees (11.3%), public-school teachers (7.1%), private-company employees (14.7%), shop and restaurant personnel (i.e., hairdresser, cook, waiter, beautician, clerk, 8.5%), and highly specialized professionals with their own activity (i.e., engineers, lawyers, graphic designers, businessmen, 13.8%).

### 2.3. Procedure

During the lockdown period of the second wave of contagions in Italy, data were collected online by administering the psychometric tools from 25 April 2020 to 10 June 2020.

Their administration was carried out in a single session using Google Forms, and the completion took about 20 min.

### 2.4. Psychometric Tools

The following data were collected for each participant: sex, age, degree of education, occupation, any chronic condition (presence/absence of, e.g., diabetes, hypertension, psychiatric pathologies), how work/study activities had changed as a result of the health emergency (interruption/online/in presence with anti-contagion measures), and how often, on average, participants left home weekly (<2/2–4/4–6/7 times).

Then, psychometric tests were used to assess levels of stress, the dispositional dimension (personality factors and intolerance to uncertainty), and the behavioral dimension (coping strategies).

More in detail, the Italian version of the Mesure du Stress Psychologique (M.S.P.) has been used to assess psychological stress [41]. It is a self-report questionnaire consisting of 49 items evaluated on a 4-point Likert scale, from 1 (not at all) to 4 (a lot). Stress is seen as a multidimensional construct that constitutes, within a certain range, an adaptive response to stimuli, while in the case of prolonged strain it becomes pathogenic. The aim of the test, therefore, is to identify dysfunctional stress by measuring different factors: the loss of control and irritability, psychophysiological sensations (accelerated heart rate, hard breathing, muscle tension, variable temperature), the sense of effort and confusion, depressive anxiety, pain and physical problems, hyperactivity and accelerated behaviors (walking, eating). Except for four statements (22, 24, 43, and 49), the items describe dysfunctional stress-related responses, i.e.: “I am irritable, my nerves are on edge, I lose patience with people and things”, and different cut-offs are identified to assess the severity of symptoms. In the Italian validation, the average M.S.P. scores for distinct subgroups by sex were M = 89.01 for men and M = 91.85 for women, but the difference was found to be not significant [41]. Cronbach’s alpha coefficient, in our study, was equal to 0.9.

The Italian version of the Ten-Item Personality Inventory (TIPI) [42] was used to assess personality traits (extraversion; agreeableness; conscientiousness; neuroticism; openness) according to the Big Five theory. It is composed of ten items, and each item consists of two descriptors separated by a comma. At the beginning of the instrument, the phrase “I see myself as…” is presented so that individuals can respond to the items based on this statement. The items are on a seven-point Likert scale ranging from 1 (strongly disagree) to 7 (strongly agree). The internal consistency of this Italian version of the Ten-Item Personality Inventory (TIPI) showed good values in our study (extraversion = 0.82; agreeableness = 0.78; conscientiousness = 0.79; neuroticism = 0.71; openness = 0.74).

The Italian version of the Intolerance of Uncertainty Scale—Revised (IUS-R) [43] has been administered to assess intolerance to uncertainty, which refers to a set of negative beliefs about uncertainty and its consequences; it is thought to play a key role in the development and maintenance of pathological worry. The IUS-R is composed of 12 items on a 5-point Likert scale from 1 (strongly disagree) to 5 (strongly agree). Regarding the total score on the IUS-R, Cronbach’s alpha coefficient was found to be 0.87 in our study.

The Coping Orientation to Problems Experienced—New Italian Version (Cope-NVI) [44], a questionnaire consisting of 60 items, has been administered to assess the coping style of the sample. The questionnaire asks individuals to rate the frequency with which they engage in specific coping processes in difficult or stressful situations. The response options range from “I usually don’t do it” to “I almost always do it”. The instructions emphasize that the subject should not refer to a specific stressor but rather think about how they typically behave in stressful situations. The questionnaire investigates five factors, which are listed here with the Cronbach’s alpha found in our study: social support (items refer to seeking understanding, information, and emotional relief; alpha = 0.72); avoidance strategies (a diverse scale including denial, substance use, behavioral and mental detachment; alpha = 0.84); positive attitude (acceptance, containment, and positive reinterpretation of events; alpha = 0.77); problem orientation (use of active strategies and planning; alpha = 0.88); transcendent orientation (items refer to religion and the absence of humor; alpha = 0.81).

### 2.5. Data Analysis

Data were analyzed using SPSS, version 27. The Cronbach’s alphas of our measures were checked. Then, descriptive statistics were calculated for each variable. Their distribution was investigated using the Shapiro–Wilk test and normal distribution was assumed with *p* > 0.05. All variables resulted as normally distributed. The homogeneity of variance was estimated with Levene’s test. In cases of heteroscedasticity, Welch’s correction was applied. A Z-test was used to analyze the difference between a sample’s IUS-R mean score and Italian population’s one, while independent *t*-tests and one-way ANOVA were used for estimating mean differences between two or more groups with relevant effect-size indexes (Cohen’s d and η^2^, respectively). In the case of multiple comparisons, Bonferroni’s correction was implemented. Pearson’s correlations were computed for exploring systematic relationships among measures, and multiple linear regression was used to estimate the predictors of stress levels.

## 3. Results

### 3.1. Descriptive Statistics

Preliminarily, minimum, maximum, mean scores (M), and standard deviations (SD) were calculated for each test and its related dimensions (Table 1).

The mean level of stress (M = 94.31) was not above the critical threshold proposed by the manual (cut-off: 104.5). However, the standard deviation suggests that several subjects showed concerning levels of stress (SD = 26.61).

Openness seems to be the most present personality trait in the sample (M = 10.09; SD = 1.86), followed by agreeableness (M = 9.15; SD = 1.91), conscientiousness (M = 8.86; SD = 1.89), neuroticism (M = 8.84; SD = 1.91), and extraversion (M = 8.45; SD = 1.96).

The mean level of intolerance to uncertainty (M = 32.16; SD = 9.59), instead, was significantly above the mean score found by Bottesi et al. [43] before the COVID-19 pandemic in the Italian context (M = 26.73; SD = 8.2; Z = 17.6; *p* < 0.01). In this case, a Z-test was performed, as validated IUS-R cut-offs were not available.

Analyzing the coping strategies, it appears that the most-used were, in order, positive attitude (M = 35.53; SD = 5.51), social support (M = 29.91; SD = 8.23), problem orientation (M = 29.65; SD = 6.06), avoidance strategies (M = 26.75; SD = 5.51), and transcendent orientation (M = 21.09; SD = 5.48).

### 3.2. Mean Differences

Independent *t*-tests and one-way ANOVA allowed to explore the differences in stress levels, dispositional dimensions (personality factors and intolerance to uncertainty), and behavioral dimensions (coping strategies) between sex, educational, professional, health, pandemic-related work schedule (interruption/online/in presence), and outdoor routine (<2/2–4/4–6/7 weekly occasions to go out) groups.

#### 3.2.1. Differences in Stress Levels

Focusing on stress levels, they affected more women (M = 96.79; SD = 27.15; *t*(279) = −4; *p* < 0.01; d = −0.37) than men (M = 87.2; SD = 23.7). Regarding differences between categories of schooling, after Bonferroni’s correction, each significance was lost. When different professionals were explored, significant differences were found (*F*(6, 543) = 5.50; *p* < 0.01; η^2^ = 0.06). After Bonferroni’s correction, health professionals were significantly less stressed (M = 84.3; SD = 21.2) than students/trainees (M = 99.9; SD = 26.1; *t*(543) = −4.24; *p* < 0.001; d = −0.6), and than the unemployed/housewives/retirees (M = 100.6; SD = 29.6; *t*(543) = −3.58; *p* < 0.01; d = −0.62). Students/trainees, instead, were significantly more stressed (M = 99.9; SD = 26.1) than private company employees (M = 88.4; SD = 25.7; *t*(543) = 3.29; *p* < 0.05; d = 0.44), and highly specialized professionals with their own activity (M = 87.6; SD = 24.1; *t*(543) = 3.45; *p* < 0.05; d = 0.47). Moreover, participants with chronic pathologies were more stressed (M = 101; SD = 30.4) than the healthy ones (M = 92.8; SD = 25.5; *t*(125) = −2.52; *p* < 0.05; d = −0.29).

Stress levels varied significantly according to the changes in work activity (*F*(2, 531) = 7.25; *p* < 0.001; η^2^ = 0.03). After Bonferroni’s correction, those who stopped working were significantly more stressed (M = 96.9; SD = 27.8) than those who kept working in presence with anti-contagion protections (M = 84.4; SD = 22.6; *t*(531) = 3.63; *p* < 0.001; d = 0.48). Interestingly, those who kept working online were also more stressed (M = 95.2; SD = 26) than those who kept working in presence with anti-contagion protections (*t*(531) = 3.38; *p* < 0.01; d = 0.42). Significant differences were found also according to the frequency of time spent outdoors weekly (*F*(3, 546) = 6.76; *p* < 0.001; η^2^ = 0.036). After Bonferroni’s correction, those who left home less than twice a week were more stressed (M = 97.5; SD = 26.9) than those who left home between two and four times (M = 87.8; SD = 25.5; *t*(546) = 2.94; *p* < 0.05; d = 0.36), those who left home between four and six times (M = 83.2; SD = 23.7; *t*(546) = 2.77; *p* < 0.05; d = 0.54), and those who left home at least once a day (M = 86.3; SD = 23.3; *t*(546) = 2.9; *p* < 0.05; d = 0.42). No significant differences were found between these last three groups.

#### 3.2.2. Differences in Dispositional Dimensions

Women showed higher levels of agreeableness (M = 9.26; SD = 1.93) and neuroticism (M = 8.95; SD = 1.97) than men (agreeableness: M = 8.83; SD = 1.83; *t*(548) = −2.32; *p* < 0.05; d = −0.22; neuroticism: M = 8.51; SD = 1.74; *t*(548) = −2.35; *p* < 0.05; d = −0.23). No significant differences were found between levels of schooling, professions, health conditions, changes in work routine, and the number of weekly occasions to go out.

By exploring intolerance to uncertainty, it has been noted that females were significantly more intolerant (M = 32.7; SD = 10) than males (M = 30.6; SD = 8.03; *t*(304) = −2.53; *p* < 0.05; d = −0.23). No significant differences were found by considering levels of schooling. Regarding professions, a general significance was detected (*F*(6, 543) = 3.14; *p* < 0.01; η^2^ = 0.033). However, after Bonferroni’s correction, only health workers appeared significantly less intolerant to uncertainty (M = 28.3; SD = 7.53) than students/trainees (M = 33.8; SD = 9.68; *t*(543) = −4.04; *p* < 0.001; d = −0.57). No significant differences based on health conditions and changes in work routine were found. Instead, a significance was detected in the frequency of time spent outdoors weekly (*F*(3, 546) = 3.16; *p* < 0.05; η^2^ = 0.017). After Bonferroni’s correction, whose who left home less than twice a week were found to be more intolerant to uncertainty (M = 32.9; SD = 9.89) than those who left home between two and four times (M = 29.5; SD = 8.77; *t*(546) = 2.85; *p* < 0.05; d = 0.35).

#### 3.2.3. Differences in Behavioral Dimensions

Regarding coping strategies, women resorted more to seeking social support (M = 30.8; SD = 8.02) and transcendent orientation (M = 22; SD = 5.31) than men (social support: M = 27.2; SD = 8.28; *t*(548) = −4.56; *p* < 0.001; d = −0.44; transcendent orientation: M = 18.5; SD = 5.19; *t*(548) = −6.68; *p* < 0.001; d = −0.65). Men, instead, were more prone to problem orientation (M = 30.6; SD = 6.54) than women (M = 29.3; SD = 5.86; *t*(548) = 2.18; *p* < 0.05; d = 0.21). The level of schooling significantly affected avoidance strategies (*F*(3, 546) = 5.18; *p* < 0.01; η^2^ = 0.028), problem orientation (*F*(3, 546) = 2.75; *p* < 0.05; η^2^ = 0.015), and transcendent orientation (*F*(3, 546) = 2.78; *p* < 0.05; η^2^ = 0.015). After Bonferroni’s correction, it was found that those who held a middle school diploma used more avoidance strategies (M = 28; SD = 5.05) than those who attended a postgraduate course (M = 23.8; SD = 4.69; *t*(546) = 3.03; *p* < 0.05; d = 0.76), and even those who held a high school diploma relied more on avoidance (M = 27.4; SD = 5.29) than those who attended a postgraduate course (*t*(546) = 3.57; *p* < 0.01; d = 0.65). Significant differences in problem orientation, instead, were lost after Bonferroni’s correction. In the case of transcendent orientation, those who had a high school diploma used this strategy more (M = 21.8; SD = 5.64) than those who had a university degree (M = 20.4; SD = 5.48; *t*(546) = 2.69; *p* < 0.05; d = 0.24). General significant differences were found between professions for avoidance strategies (*F*(6, 543) = 3.24; *p* < 0.01; η^2^ = 0.035) and transcendent orientation (*F*(6, 543) = 3.17; *p* < 0.01; η^2^ = 0.034). However, after Bonferroni’s correction, it resulted that only health professionals used less avoidance strategies (M = 24.9; SD = 5.37) than the unemployed/housewives/retirees (M = 28.4; SD = 5.3; *t*(543) = −3.61; *p* < 0.01; d = −0.63) and shop/restaurant personnel (M = 28.1; SD = 5.15; *t*(543) = −3.07; *p* < 0.05; d = −0.58). Moreover, those who suffered from a chronic pathology used transcendent orientation significantly more (M = 22.5; SD = 5.64) than healthy participants (M = 20.8; SD = 5.41; *t*(548) = −2.8; *p* < 0.01; d = −0.31).

Changes in the work routine affected the search for social support (*F*(2, 531) = 8.05; *p* < 0.001; η^2^ = 0.029) and avoidance strategies (*F*(2, 531) = 10; *p* < 0.001; η^2^ = 0.036). After Bonferroni’s correction, it was found that those who stopped working looked for social support (M = 31.6; SD = 7.92) more than those who kept working in presence with anti-contagion measures (M = 27.4; SD = 7.79; *t*(531) = 3.96; *p* < 0.001; d = 0.52), and used more avoidance strategies (M = 28.2; SD = 5.62) than those who kept working online (M = 26.2; SD = 5.42; *t*(531) = 3.83; *p* < 0.001; d = 0.37) and those who kept working with anti-contagion measures (M = 25.5; SD = 5.04; *t*(531) = 3.83; *p* < 0.001; d = 0.51). Significant differences were found also in social support (*F*(3, 546) = 3.25; *p* < 0.05; η^2^ = 0.018) regarding how much time was spent outdoors weekly. After Bonferroni’s correction, it was found that those who left home less than twice a week looked for social support (M = 30.5; SD = 8.22) more than those who went out at least once a day (M = 27.2; SD = 7.97; *t*(546) = 2.75; *p* < 0.05; d = 0.4).

### 3.3. Pearson’s Correlations

Pearson’s correlations were calculated for exploring systematic relationships between variables.

First of all, age correlated positively with transcendent orientation (r = 0.26; *p* < 0.001), problem orientation (r = 0.15; *p* < 0.001), and avoidance strategies (r = 0.08; *p* < 0.05), while it correlated negatively with stress (r = −0.12; *p* < 0.01), agreeableness (r = 0.1; *p* < 0.05), intolerance to uncertainty (r = 0.1; *p* < 0.05), and seeking social support (r = −0.15; *p* < 0.001).

The Pearson’s correlations between stress and dispositional dimensions are shown in Table 2, while the Pearson’s correlations between stress and behavioral dimensions are shown in Table 3, and the Pearson’s correlations between dispositional and behavioral dimensions are shown in Table 4.

Those who suffered from higher levels of stress were also significantly more intolerant to uncertainty, less open to new experiences, more prone to respect rules, and more interested in building social relationships.

Those who suffered from higher levels of stress used avoidance strategies and social support significantly more, while they showed lower positive attitude and transcendent orientation.

Those who had higher levels of extraversion maintained higher levels of positive attitude, while those who had higher levels of agreeableness and conscientiousness used more social support and avoidance strategies. People with higher scores of neuroticism kept a transcendent orientation and sought more social support. People with higher levels of openness used less avoidance strategies, kept a more positive attitude, and were more focused on a transcendent orientation and on the problem. Finally, those who were more intolerant to uncertainty used more avoidance strategies, social support, and transcendent orientation, and kept less of a positive attitude.

### 3.4. Multiple Linear Regression

To conclude, significant predictors of stress levels were explored among dispositional (extraversion, agreeableness, conscientiousness, neuroticism, openness, intolerance to uncertainty) and behavioral (social support, avoidance strategies, positive attitude, problem orientation, transcendent orientation) variables (Table 5).

The model showed a R^2^ equal to 0.40, with the only significant predictors found in intolerance to uncertainty (β = 0.4, t = 10.4, *p* < 0.001), avoidance strategies (β = 0.23, t = 6.17, *p* < 0.001), social support (β = 0.16, t = 4.24, *p* < 0.001), positive attitude (β = −0.12, t = −3.3, *p* < 0.001), and openness (β = −0.11, t = −2.9, *p* < 0.01).

## 4. Discussion

The global COVID-19 pandemic caused substantial disruptions of daily routines, leading to notable effects on individuals’ mental well-being, as discussed previously. Additionally, given the rapid transmission of the virus, it is understandable that heightened concerns and anxiety have emerged among the general population [45].

This study aimed at understanding how stress was affected by dispositional and behavioral dimensions and which categories of the population were most hit by changes of the health emergency, in order to reflect on targeted interventions inspired by health psychology. Although our sample showed a mean level of stress under the critical threshold of the M.S.P., the standard deviation pointed out a highly dispersed subjective experience, with some individuals reporting low or normal amounts of strain, and others struggling with severe stress. This further supports our literature-based hypothesis of important differences between persons, which distinguish between those who maintained an adaptive level of stress and those who suffered from it on par with a clinical syndrome. For this reason, we proceeded with our analyses to uncover such differences and to identify the intervening factors, in order to understand how to reduce and maintain stress within an adaptive range. We took into account many variables; therefore, we started our analysis with a concise, but rigorous and straight-to-the-point model based on differences between groups.

The study found that stress primarily affected women and vulnerable individuals, particularly students/trainees, those with chronic diseases, people who stopped working because of restrictions, and those who left home less than twice a week. These data are in agreement with previous scientific evidence. In fact, with the pandemic’s progression, students and females showed greater levels of stress, depression, and anxiety that needed specific interventions for cognitive, affective, and psychosocial symptoms [46,47,48]. It should also be noted that our investigation highlighted a significant sex difference regarding stress levels, while in the validation study of the M.S.P. conducted in Italy before COVID-19, this significance was not found. This further supports the idea that the emergency had a different impact between men and women, amplifying the divide between the sexes. Moreover, scholars found that unemployed individuals [49] and psychiatric patients [50,51] were more prone to be distressed. This is particularly compelling when special needs must be taken into account [52]. According to other studies, a higher dietary intake of vitamin D and sunlight exposure were associated with a lower likelihood of having high perceived stress among physically active individuals, so even the conditions of lockdown could have worsened the health status because of sedentary habits: in fact, going out frequently and performing regular physical activity offers many positive physiological and psychological advantages [53].

It experiences interesting that this study highlighted that healthcare professionals experienced lower levels of stress when compared to other categories. This could be due to their continued employment and exemption from work closures and suspensions. This finding might seem in contrast with most of scientific literature evidence. Indeed, according to a systematic review [54], healthcare workers showed a greater level of stress that resulted in a higher risk of post-traumatic stress disorder (PTSD). In particular, youth, a low work-experience, female sex, unsafe settings, a lack of training, and a lack of social support seemed to be the risk factors for this condition [55].

In our study, lower levels of stress in healthcare workers did not mean the absence of stress, but it is a finding that should be contextualized within the comparison between groups. We can hypothesize that belonging to a healthcare profession played a dual role: protecting individuals from the initial impact of the pandemic by maintaining employment, and also negatively burdening them with time. There is another explanation that should be taken into account. The M.S.P. measures the personal perception of stress, so an individual must be aware of their own level of tension in order to answer the questionnaire accurately. To develop such awareness, one should have the time to reflect and process the burden they are bearing. Focusing on healthcare professionals, they found themselves thrown into the emergency probably without the time to ponder what they were facing. This could have biased their responses to the M.S.P., and led them to successive burnout and mental health troubles.

Healthcare professionals, moreover, demonstrated higher levels of tolerance of uncertainty compared to students/trainees, while women and those who spent time outdoors less than twice a week were more intolerant. The ability to manage uncertainty was particularly lacking among vulnerable populations, especially those still in the process of education. It should be remarked that, according to the literature, those with higher intolerance to uncertainty also experienced higher levels of stress [56], neuroticism, extraversion, and conscientiousness [16].

Regarding the results on coping strategies, it was highlighted that women were more oriented towards seeking social support and transcendent orientation, while men referred more to problem-oriented coping. These data differ slightly from other studies [57], which highlighted that males tend more towards avoidance than females. Moreover, those with a middle school diploma made significantly greater use of avoidance strategies, and those with a high school diploma were more prone to avoidance and transcendent orientation, when both groups were compared to postgraduate courses. This is in line with the literature [57]. Even the unemployed/housewives/retired, the personnel of shops and restaurants, and those who stopped working during the lockdown resorted more to avoidance. As they were also the most affected by pandemic events, they could have felt incapable of dealing with COVID-19 and tried to avoid the critical issues. Moreover, those who stopped working and those who left home less than twice a week sought more social support, and those with chronic pathologies, instead, relied on transcendent orientation.

Apart from statistical significance, the effect size should be taken into account for the interpretation of results. On average, in our *t*-tests and one-way ANOVA, we found medium effect sizes. In more detail, taking into account specific comparisons, the highest effect size was detected in schooling differences in the use of avoidance strategies, bordering a large effect size (for instance, in the case of middle school compared to postgraduate courses: d = 0.76), followed by sex differences in transcendent orientation (i.e., women compared to men: d = −0.65). This means that our findings corresponded to a real practical impact, especially when referring to coping strategies, stress, and intolerance to uncertainty.

Correlations with age showed that younger people were more distressed, agreeable, intolerant to uncertainty, and focused on social support. By contrast, older people resorted more to transcendent orientation, followed by problem orientation and avoidance strategies. This finding can be explained by past research, which showed that the older population, being at greater risk of death and infirmity for COVID-19, developed greater anxiety and depression [58,59] and took refuge in their religious beliefs. Proactive coping, however, was also a resilience strategy for COVID-19 in older adults [58].

Agreeableness and conscientiousness were positively associated with stress [60]. This could be explained as due to lockdown measures negatively affecting social relationships and amplifying the adherence to strict rules [61]. Our results also highlight a strong correlation between intolerance of uncertainty and stress. These data are perfectly in line with the literature [62,63]: as COVID-19 was a new virus, it inevitably brought ambiguity. Greater stress, therefore, suggests greater intolerance to unpredictability, risk, anxiety, and fear [64]. A negative relationship, instead, was found between stress and openness.

Regarding coping strategies used to manage stress, seeking social support and trying to avoid negative thoughts worsened stress levels, whereas a positive attitude and a transcendent orientation reduced them. With regard to social support, there are several pieces of evidence that contradict our results. Indeed, emotion-focused coping strategies (such as seeking social support and acceptance) correlated negatively with perceived stress, whereas a dysfunctional coping style (i.e., the avoidance of negative thoughts, denial, and substance use) correlated positively with it [65,66]. These mixed results can be attributed to the unprecedented emergency of the pandemic, which challenged traditional means of finding comfort in the population. The search for support in other people, indeed, could have led to the paradoxical effect of fostering the contagion of uncertainty and anxiety. Only those who maintained a positive attitude and those who resorted to religion to find an answer were able to reduce their stress levels. These last results are consistent with the literature [65,66].

From the correlational data, relationships also emerged between personality factors and coping strategies. In particular, it was noted that extroverts had higher levels of positive attitude, while those who were more agreeable, conscientious, and intolerant to uncertainty resorted more to social support and avoidance strategies. Participants with higher levels of neuroticism, on the other hand, tended to seek social support more and maintained a transcendent orientation. These data differ slightly from those of Gashi and colleagues, who instead highlighted that those with high levels of extraversion and neuroticism maintained coping styles based on social support and avoidance [67]. People with higher levels of openness used fewer avoidance strategies, adopting a coping based on a positive attitude, problem analysis, and transcendent orientation. These results are in line with other studies [57], as it has been found that people with an open mind see problems as challenges to deal with, often resorting to remedies such as hobbies and charitable activities, with the aim of expanding their experiential baggage [68]. Participants more intolerant to uncertainty also kept a lower positive attitude and a higher transcendent orientation.

To conclude, we verified whether stress levels were predicted by either dispositional or behavioral factors. According to our multiple linear regression model, both dimensions were significantly involved, but stress was mainly affected by behavioral factors. In particular, significant predictors were intolerance to uncertainty, avoidance strategies, social support, positive attitude, and openness. This supports the idea that during the lockdown, stress needed to be managed with both behavioral interventions focused on coping styles and modification strategies of people’s attitudes and beliefs.

Reflecting on our results, young people could benefit from counselling and mindfulness, even via a chatbot [69], while those who lost their job or those with low schooling could be supported with career service consulting [70]. In addition, the population could be guided towards at least 150 min a week of physical activity, even with online coaching, and could be involved in groups of mutual support [71] for depressive symptoms [63,64]. Moreover, stress levels could be reduced via interventions promoting a perception of control and a new perspective on the ongoing events, as well as the acceptance of difficulties, the maintenance of a positive attitude, and the search for constructive social support.

### Limitations

Our findings should be considered in light of some limitations. In particular, the sampling procedure was conducted with a snowball method and our participants were heterogeneous. Considering the period of lockdown when our study was conducted, the use of online surveys was the only safe and allowed means of the collection of data. While this may have biased our findings, it allowed us to work on first-hand evidence on the ongoing psychological status of the population. Moreover, including several demographic groups was part of our scope. We managed the bias of such variability by subdividing the sample into subparts and treating them separately, coherently with our objectives. However, we acknowledge the risks of such variability and we hope that future studies will deepen this topic for an increased rigorous understanding of pandemic-related phenomena.

## 5. Conclusions

Research produced in the last years represents an important source of knowledge and awareness in the case of other emergencies. The large amount of literature produced on COVID-19 should now be reviewed for designing targeted interventions. These strategies, however, should not be the same for everyone.

The results of such work would remain at the disposal of the community, for a prompt evidence-based and targeted management of health risks. The final general goal is to support the entire world population, and in particular those who are the most fragile.

## Figures and Tables

**Table 1 ijerph-21-00330-t001:** Descriptive statistics of the psychometric tests.

Measure	Mean	SD	Minimum	Maximum
Stress	94.31	26.61	54	184
Extraversion	8.45	1.96	2	14
Agreeableness	9.15	1.91	2	14
Conscientiousness	8.86	1.89	2	14
Neuroticism	8.84	1.92	2	14
Openness	10.09	1.86	2	14
Intolerance to uncertainty	32.16	9.59	13	60
Social support	29.91	8.23	12	48
Avoidance strategies	26.75	5.51	16	50
Positive attitude	35.53	5.51	16	48
Problem orientation	29.65	6.06	12	44
Transcendent orientation	21.09	5.48	8	32

**Table 2 ijerph-21-00330-t002:** Pearson’s correlations between stress and dispositional dimensions.

	Extraversion	Agreeableness	Conscientiousness	Neuroticism	Openness	Intolerance to Uncertainty
Stress	−0.08	0.12 **	0.14 **	0.06	−0.17 **	0.55 **

** *p* < 0.01.

**Table 3 ijerph-21-00330-t003:** Pearson’s correlations between stress and behavioral dimensions.

	Social Support	Avoidance Strategies	Positive Attitude	Problem Orientation	Transcendent Orientation
Stress	0.3 **	0.43 **	−0.16 **	−0.07	−0.1 *

* *p* < 0.05; ** *p* < 0.01.

**Table 4 ijerph-21-00330-t004:** Pearson’s correlations between dispositional and behavioral dimensions.

	Social Support	Avoidance Strategies	Positive Attitude	Problem Orientation	Transcendent Orientation
Extraversion	0.02	0.02	0.15 **	0.04	−0.05
Agreeableness	0.23 **	0.16 **	0.02	−0.05	−0.03
Conscientiousness	0.17 **	0.15 **	0.01	−0.04	−0.06
Neuroticism	0.14 **	0.05	−0.02	0.02	0.15 **
Openness	0.03	−0.1 *	0.16 **	0.08 *	0.09 *
Intolerance to uncertainty	0.2 **	0.39 **	−0.1 *	−0.03	0.17 **

* *p* < 0.05; ** *p* < 0.01.

**Table 5 ijerph-21-00330-t005:** Multiple linear regression: predictors of stress levels (M.S.P.).

Predictors	t	*p*	β
Extraversion	−0.05	0.9	−0.001
Agreeableness	0.07	0.9	0.002
Conscientiousness	1.01	0.3	0.03
Neuroticism	−1.58	0.1	−0.05
Openness	−2.9	0.004	−0.11
Intolerance to uncertainty	10.4	<0.001	0.4
Social support	4.24	<0.001	0.16
Avoidance strategies	6.17	<0.001	0.23
Positive attitude	−3.3	<0.001	−0.12
Problem orientation	−1.52	0.1	−0.06
Transcendent orientation	0.9	0.9	0.03

## Data Availability

The data presented in this study are available upon request from the corresponding author. The data are not publicly available due to reasons related to the protection of the privacy of the participants.

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
