# Peer review of "Impact of Stress during COVID-19 Pandemic in Italy: A Study on Dispositional and Behavioral Dimensions for Supporting Evidence-Based Targeted Strategies"

_ijerph, 2024, doi:10.3390/ijerph21030330_

Round 1
Reviewer 1 Report
Comments and Suggestions for Authors
This study is well-written and well-presented. I only have one primary concern for this study. For the inferential statistical analysis; the authors need to add effect size analysis and not only depend on the p-value. The authors can read more about this in articles discussing the misuse of p-values. Of course, this needs to be further addressed in the discussion section.
Author Response
This study is well-written and well-presented.
We thank the Reviewer for their kind comment on our manuscript.
I only have one primary concern for this study. For the inferential statistical analysis; the authors need to add effect size analysis and not only depend on the p-value. The authors can read more about this in articles discussing the misuse of p-values. Of course, this needs to be further addressed in the discussion section.
We thank the Reviewer for their valuable suggestion. We added the effect-sizes in all analyses in order to better understand the impact of our results. In more detail, both Cohen’s d and eta2 have been used. As suggested, this point has been addressed in the Discussion section from line 412.
Reviewer 2 Report
Comments and Suggestions for Authors What is the "question" of this paper? The author himself has not been able to answer the question "But why still talk of COVID-19 pandemic?" on line 91. From line 92 onwards, it says that people's vulnerability to damage varies depending on their position, which is why this paper intended to address this issue. However, the author's ``conceptions'' are written without citing any papers. In fact, a large body of research has been produced since the early days of the pandemic showing that the toll of the pandemic has been unequal. Many tests are repeated in "Mean differences" from line 211 onwards. In this case, the standard for significant difference is not 5%, but the p value must be changed depending on the number of tests. Methods to change the p-value include the Bonferroni method, Holm method, and Benjamini & Hochberg method. There are also problems with the way statistical results are written. For example, "the youngest (age group: 18-24, M = 89,2; SD = 27,06; F = 2,03)" in line 216. The numbers in parentheses indicate the results of the ANOVA, that is, whether there was a difference between the age groups. A post-hoc test is required to show that "18-24" was significantly lower. The results of "3.4. Multiple linear regression" from line 272 onwards should be tabulated. Many documents that appear for the first time are cited in "Conclusion" starting from line 400 (References 60 to 65). The Conclusion is a part that summarizes the entire paper and is not a part to expand the discussion.Author Response
What is the "question" of this paper?
We thank the Reviewer for their question. We worked further on the logical coherence between objectives – analyses – results – discussion of our manuscript, in order to outline better the research questions of this paper. Therefore, we re-wrote clearer the objectives (lines: 109 – 115) and we adjusted all the following paragraphs to address and respond precisely to them. All changes were maintained in revision mode.
The author himself has not been able to answer the question "But why still talk of COVID-19 pandemic?" on line 91. From line 92 onwards, it says that people's vulnerability to damage varies depending on their position, which is why this paper intended to address this issue. However, the author's ``conceptions'' are written without citing any papers. In fact, a large body of research has been produced since the early days of the pandemic showing that the toll of the pandemic has been unequal.
We thank the Reviewer for their comment. We argued better the rationale of our work and above all we supported with the literature our conceptions (lines 92 – 103). COVID-19 has been a dramatic experience where stress affected unevenly the population, therefore wanted to discuss our data to understand dispositional and behavioral components of such burden, taking also into account group differences for targeted strategies. Your comment helped us giving support to our premises and making more recognizable our general goal.
Many tests are repeated in "Mean differences" from line 211 onwards. In this case, the standard for significant difference is not 5%, but the p value must be changed depending on the number of tests. Methods to change the p-value include the Bonferroni method, Holm method, and Benjamini & Hochberg method.
We thank the Reviewer for their precious correction. We ran Bonferroni’s method to mean differences. This brought some changes in our results. However, the main findings of the manuscript were confirmed. Results have been rewritten from line 204 onward, and Abstract and Discussions have been adjusted accordingly. We improved the paragraph on Sample composition and we tabulated correlations to make them clearer in relation to the manuscript rationale (lines 326 – 336). We also ran an extensive English revision. Therefore, your prompt was precious to improve the general coherence of statistics and results.
There are also problems with the way statistical results are written. For example, "the youngest (age group: 18-24, M = 89,2; SD = 27,06; F = 2,03)" in line 216. The numbers in parentheses indicate the results of the ANOVA, that is, whether there was a difference between the age groups. A post-hoc test is required to show that "18-24" was significantly lower.
We thank the Reviewer for their comment. All the results have been re-written according to the international guidelines for reporting statistical analyses.
The results of "3.4. Multiple linear regression" from line 272 onwards should be tabulated.
Many thanks for their comment. We tabulated the results or Multiple Linear Regression (from line 351).
Many documents that appear for the first time are cited in "Conclusion" starting from line 400 (References 60 to 65). The Conclusion is a part that summarizes the entire paper and is not a part to expand the discussion.
Many thanks for the comment. We moved all the literature in the Discussion section, and Conclusion was dedicated to summarizing the paper only.